# OPENCITY: A SCALABLE PLATFORM TO SIMULATE URBAN ACTIVITIES WITH MASSIVE LLM AGENTS

## ABSTRACT

Agent-based models (ABMs) have long been employed to explore how individual behaviors aggregate into complex societal phenomena in urban space. Unlike black-box predictive models, ABMs excel at explaining the micro-macro linkages that drive such emergent behaviors. The recent rise of Large Language Models (LLMs) has led to the development of LLM agents capable of simulating urban activities with unprecedented realism. However, the extreme high computational cost of LLMs presents significant challenges for scaling up the simulations of LLM agents. To address this problem, we propose OpenCity, a scalable simulation platform optimized for both system and prompt efficiencies. Specifically, we propose a LLM request scheduler to reduce communication overhead by parallelizing requests through IO multiplexing. Besides, we deisgn a "group-and-distill" prompt optimization strategy minimizes redundancy by clustering agents with similar static attributes. Through experiments on six global cities, OpenCity achieves a 600-fold acceleration in simulation time per agent, a 70% reduction in LLM requests, and a 50% reduction in token usage. These improvements enable the simulation of 10,000 agents' daily activities in 1 hour on commodity hardware. Besides, the substantial speedup of OpenCity allow we to establish a urban simulation benchmark for LLM agents for the first time, comparing simulated urban activities with real-world data in 6 major cities around the world. We believe our OpenCity platform provides a critical infrastructure to harness the power of LLMs for interdisciplinary studies in urban space, fostering the collective efforts of broader research communities. Code repo is available at https://anonymous.4open.science/r/Anonymous-OpenCity-42BD.

## 1 INTRODUCTION

Agent-based models (ABMs) were first introduced to urban studies in the seminal work of Thomas Schelling about 50 years ago Schelling (2006), which ingeniously explained how segregation can emerge as the aggregation of individual choices. Compared to black-box predictive models, ABMs offer the unique advantage of explaining the underlying mechanisms behind aggregated phenomena, *i.e.*, revealing the connections between "micro-motives" and "macro-behaviours." As a result, ABMs play an important role in many research areas, including computational social sciences, urban planning and public health. The recent advance of Large Language Models (LLMs) have driven the rise of LLM agents Park et al. (2023); Xu et al. (2023), which leverage LLM's remarkable capabilities of commonsense reasoning and role-playing to simulate human behaviours. Unlike previous rule-based agents, these emerging LLM agents generate far more realistic human behaviours Park et al. (2023); Shao et al. (2024), and can also explain their inner motives via prompting techniques like chain-of-thoughts Wei et al. (2022). Therefore, LLM agents hold great potential to harness the power of language models in transforming urban studies.

Despite this promising outlook, LLM agents also face severe challenges of scaling up due to the high computation time. In the pioneering work of Park et al. Park et al. (2023) only 15 LLM agents were employed to simulate a small village. One main reason is the prohibitive simulation time, which can be broken down into two parts: on one hand, LLMs are inherently slow due to their enormous model sizes; on the other hand, powerful commercial LLMs are only accessible via APIs, which introduces significant time delay due to network transmission, further slowing down simulation. To make matters worse, the prompt design of urban LLM agents often involve dynamic elements, such

as the changing memories and perceived environment Park et al. (2023); Shao et al. (2024). This important feature prevents the straightforward reuse of simulated behaviors from a small sample of the population Chopra et al. (2024), as LLM agents need to maintain independent memories and experiences, which are essential for simulating a vibrant and diverse urban population.

In this paper, we present OpenCity, a scalable platform that introduces both system-level and prompt-level optimizations to enable efficient LLM agent simulation in urban environments. Specifically, we design an LLM request scheduler that leverages the scalable I/O event notification mechanism in operating system (*e.g.*, epoll in Linux Bruguera i Moriscot (2019)) to minimize network transmission delay. This design is based on our key observation that sending LLM requests and receiving generated output account for only a small portion of total communication time, while the rest are wasted on waiting for LLM responses and the repeatedly establishing TCP connections Peterson & Davie (2007). To address this problem, the LLM request scheduler uses the scalable I/O event notification mechanism to parallelize LLM requests by reusing the network I/O portal and TCP connections while waiting for responses. Besides, LLM request scheduler also analyzes the interdependencies of LLM requests and local computation tasks, *e.g.*, updating agent's memory and retrieving nearby locations, ensuring local computation tasks are optimally distributed across multiple CPU cores. These system-level optimizations enable large-scale LLM agent simulations to run on commodity hardware. As for the prompt-level optimization, OpenCity introduces a novel "group-and-distill" prompt strategy to minimize the input token required by LLMs. The key idea is to identify the clusters of LLM agents that share semantically similar static elements, *e.g.*, age, gender and income level, and use shared context in batch prompting Cheng et al. (2023) to reduce token redundancy. Specifically, our "group-and-distill" strategy leverages the in-context learning capabilities of LLMs to implement a prototype learning workflow that automatically discovers clusters of LLM agents with semantically similar static elements for simulation. Agents within the same clusters are grouped into a batch prompt, and we design a "prompt distillation" to extract shared prefix for grouped agents. Finally, OpenCity also features an easy-to-use web portal that facilitates code-less simulation configuration and result visualization. This design minimizes the program requirement for running simulation with LLM agents, ensuring our OpenCity platform can benefit researchers from all background.

We evaluate the efficiency and faithfulness of OpenCity in simulating the urban activities of 6 cities around the world using the widely adopted Generative Agent workflow Park et al. (2023). Our experiments show OpenCity achieves an average 635x acceleration in simulations with 10,000 LLM agents. Besides, the number of requests and consumed tokens are reduced by 73.7% and 45.5%, respectively. OpenCity also shows strong scalability, with the simulation time per agent reducing from 36.25 to 0.06 seconds as the simulation size increases from 1 to 10,000 agents, demonstrating that larger simulations allow for more efficient LLM request scheduling and prompt distillation. More importantly, OpenCity also maintains high faithfulness of the simulated behaviour. Specifically, the Jensen–Shannon divergence and top-1 hit rates of our method are comparable to the standard prompting technique of batch prompting Cheng et al. (2023), and substantially surpass straightforward reusing strategy Chopra et al. (2024). Besides, the top-1 hit rate can reaches up to 96% when using powerful LLMs like GPT-4o.

The substantial simulation acceleration allows us to benchmark LLM agents' ability to replicate large-scale urban activities for the first time. We use classic evaluation measures such as the radius of gyration Gonzalez et al. (2008), origin-destination matrix Jiang et al. (2016), and segregation index Moro et al. (2021) to assess LLM agents' simulations. These are the most widely adopted metrics that characterize urban residents' activities at both individual and group levels, and across physical and social domains. Our experiments show that LLM agents perform comparably to, or better than, traditional rule-based agents like EPR Song et al. (2010). Moreover, LLM agents enable counterfactual analyses, such as evaluating experienced segregation in cities without residential segregation Massey & Denton (1988). They also allow researchers to interrogate LLM agents' motives behind their behaviors, offering valuable insights for urban policy-making.

The contribution of this paper are three-folds:

- We design a high-performance platform OpenCity that introduces system-level LLM request scheduler and prompt-level "group-and-distill" strategy to reduce LLM agent simulation time. OpenCity maintains high fidelity in simulated behaviors while achieving an average 635x acceleration.

- The substantial speedup allows us to benchmark LLM agents for reproducing urban activities for the first time.

- OpenCity provides a user friendly web portal, allowing researchers without programming background to easily configure simulation and visualize the results.

## 2 RELATED WORKS

### 2.1 LLM AGENTS

With the widespread use of large language models (LLMs) in various applications, the limitations of LLMs, e.g., unstable reasoning abilities, limited memory capacity, and lack of specialized expertise, have been exposed to the public. As one of the potential solution, LLM agents are proposed to overcome these limitations and promote the practical application of LLMs. AutoGPT Gravitas (2023) as one of the most popular LLM autonomous agent explore the potential of applying LLM to enable the autonomous planning and task-solving. After that, LLM agents Wang et al. (2024); Xi et al. (2023) have made significant progress in two directions: task-oriented agents and simulation agents. Following the first direction, researchers aim to improve LLM agent's ability to solve complex tasks. For example, lots of programming agents, such as ChatDev Qian et al. (2023), SWEAgent Yang et al. (2024), and MetaGPT Hong et al. (2023), are designed to solve the complex programming tasks. As for the second direction, generative agents Park et al. (2023) have demonstrated the potential of large models in simulating human behavior, which has been further validated in subsequent research. S3 Gao et al. (2023) explores the potential of using LLM agents to simulate the social network. CoPB Shao et al. (2024) defines a agentic workflow to simulate the mobility behaviors. RecAgent Wang et al. (2023) simulate the user behavior in the recommendation system. While these works demonstrate the potential of LLM agents, the large scale efficient simulation of generative agents becomes the critical bottleneck of further applications.

### 2.2 LLM DEPLOYMENT OPTIMIZATION

To support the efficient inference of LLMs and LLM agents, enormous works and systems Miao et al. (2023) are designed to optimize the inference efficiency of LLMs and further accelerate their practical applications. For example, Flash-attention Dao et al. (2022) is an IO-aware exact attention algorithm which uses tiling to reduce the number of memory reads/writes within GPU. AWQ Lin et al. (2024) is an activation-aware weight quantization to compress and accelerate the LLM inference. vLLM Kwon et al. (2023) proposes pagedAttention mechanism to enable highly efficient KV cache scheduler during the inference and becomes the most population open source LLM inference engine. SGLang Zheng et al. (2023) provides a flexible frontend language to enable the efficient autonomous optimization of LLM inference. While these systems are designed to process the general LLM inference, specific characteristics of generative agents especially urban generative agents are ignored which can be employed to further accelerate the inference and simulation. In this paper, we explore the potential of this direction and design the OpenCity platform.

## 3 PRELIMINARIES

### 3.1 LLM AGENTS FOR URBAN ACTIVITIES

We focus on using LLM agents to reproduce urban dynamics characterized primarily by physical mobility. Consider an urban environment $E$ containing $N$ LLM agents. The state of agent $i$ at simulation time $t$, denoted as $S_i(t) = \{s_i, m_i(t)\}$, consists of both static properties $s_i$ and dynamic properties $m_i(t)$. Static properties, like the agent's demographics, remain constant throughout the simulation, while dynamic properties, such as memory and perceived environment information, change frequently and are hard to predict. We can represent the state update of agent $i$ using a function $f$:

$$m_i(t+1) = f(s_i, E, m_i(t)); S_i(t+1) = \{s_i, m_i(t+1)\} \tag{1}$$

Here, $m_i(t+1)$ is the updated memory of agent $i$ at time $t+1$, and the function $f$ models how the agent updates its state by perceiving the urban environment $E$, reflecting on its memory $m_i(t)$,

and interacting with the LLM. The individual trajectory of agent $i$, denoted as $T_i$, describes the trajectory of the agent over time in the urban environment $E$. If the location of agent $i$ at time $t$ is represented by $L_i(t)$, which depends on its state $S_i(t)$, then the individual trace can be expressed as $T_i = \{L_i(0), L_i(1), L_i(2), ..., L_i(t_s)\}$, where $t_s$ is the the total simulation time. Along with individual mobility, we also examine the aggregated mobility features $A = \Phi(\sum_i \phi(\sum_t S_i(t)))$, such as Original-Destination (OD) matrix and income segregation index, which reflects the urban dynamics involving states of all agents.

To simulate LLM agents in the urban space, we set the initial state for the agents and environment $\{S_i(0)|i \in N\}$ and then apply the Equation 1 for each agent at every simulation step. When the number of agents increases, challenges arise mainly because of the LLM request process. LLMs are inherently slow due to their parameter size, and when using commercial LLMs accessed via APIs, response times can be further delayed, especially with poor network conditions. Some have proposed reusing the LLM response for agents can improve the efficiency Chopra et al. (2024), but it requires that the agents have the completely same state or have limit kinds of state that can be easily predicted. What's more, simply reusing the response would eliminate the independence of agents and reduce the faithfulness of the simulation results. Urban agent $i$ has dynamic memory $m_i(t)$ that evolves during the simulation. This memory $m_i(t)$ depends not only on past memory $m_i(t_h)$ but also on the current environment. Since decision-making and memory updates rely heavily on the LLM, predicting an agent's future state or finding an agent with an identical state to reuse an LLM response is difficult. Therefore, to simulate the large-scale and reliable LLM agents for urban dynamics, a simple response reuse strategy is insufficient.

## 3.2 TIME COST ANALYSIS

In light of the prevailing dominance of remote LLM service invocations in the current operational landscape of LLM agents simulation, a decomposition of the time required for a single LLM request can be undertaken, as illustrated in Fig.1(b). The first phase is the initialization and reception time for the LLM request, the second is the TCP/IP connection and destruction time between the simulation system and the LLM service provider, and the third is the data transmission and waiting time. For a single LLM request, the overhead of the first and second phases is relatively low in comparison to the third, and the core time consumption is derived from the data transmission and waiting.

The simulation of large-scale agents necessitates the issuance of a considerable number of LLM requests, which, given the presence of waiting periods, impairs the overall efficiency of the simulator. Furthermore, the system resources are not fully utilized. Consequently, the effective scheduling of LLM requests is essential for enhancing the overall utilization of system resources, which in turn improves the overall efficiency of the simulation. Furthermore, as the time required for LLM inference is directly proportional to the number of tokens contained in an LLM request, it is also important to reduce the number of tokens consumed per agent while compressing the number of requests.

From this vantage point, the present work puts forth an efficacious LLM request scheduler and a prompt distillation method, which can markedly enhance the efficiency of large-scale LLM agents simulation.

## 4 OPENCITY PLATFORM

We devise a scalable platform OpenCity to accelerate the simulation of urban LLM agents from both system- and prompt-level. The OpenCity platform aims to substantially reduce the simulation time per LLM agent while maintaining high simulation fidelity. Besides, OpenCity also provides a user-friendly web interface to facilitate the easy access of researchers from diverse background. The key designs are introduced as follows.

### 4.1 LLM REQUEST SCHEDULER

As shown in Fig.1(a), for a LLM agent, the dependency between its LLM requests—that is, the necessity for the next LLM request to be initiated after the previous one is completed—results in a constant waiting time under the condition of a fixed network environment and request content.

In contrast, for a system comprising multiple agents, there is no dependency between their LLM requests. In order to achieve asynchronous processing of multiple LLM requests, we have implemented an IO multiplexing scheme (based on epoll in Linux) which eliminates waiting time in the simulation system. This allows the operating system to manage IO waiting, thereby achieving the desired "zero-awareness" of data transmission in the simulation system. Consequently, the average time for a LLM request is reduced to the time required for the first and second phases (Time saving#1 in Fig.1(c)).

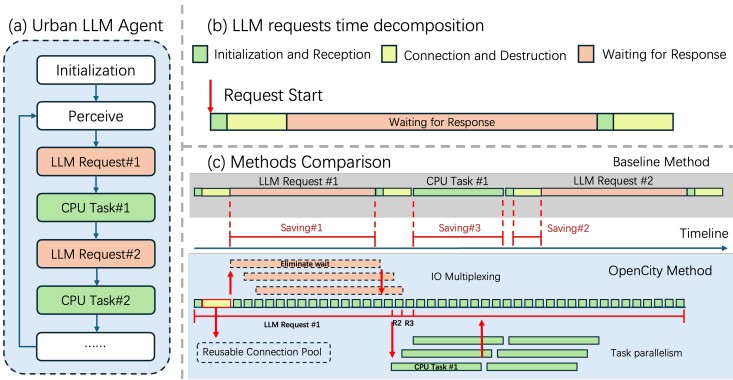

Figure 1: The functionality of the proposed LLM Request Scheduler.

Furthermore, the considerable number of LLM calls necessitates the frequent establishment of connections with the service provider, resulting in a considerable overhead in the establishment and destruction of each connection. However, given that the content of LLM requests is inherently linked to the corresponding agent, it is possible to leverage the same connection for multiple agents, thereby reducing the overall performance overhead. To address this issue, a pool of reusable connections is maintained within the system. Upon initiation of an LLM request by an agent, the request content is populated into an available connection, thus avoiding the establishment of a new connection. This approach additionally reduces the mean time consumption of LLM requests (Time saving#2 in Fig.1(c)).

For those agents with CPU tasks during the computation process, it is important to note that the continued occupation of CPU resources by the computation load will inevitably result in a delay in the sending of LLM requests from subsequent agents. To mitigate the adverse effects of this issue on the system's overall performance, we categorize the CPU task as "local IO", offload it to available cores for computation through a multi-core parallel scheme, and then return the result to the designated agent upon completion of the computation. This approach further ensures the stable operation of asynchronous LLM requests (Time saving#3 in Fig.1(c)).

The proposed LLM request scheduler is designed to reduce the waiting time for a significant number of LLM requests during the simulation runtime. Based on the supporting auxiliary scheme, it has the potential to significantly enhance the efficiency of large-scale LLM agents.

## 4.2 GROUP-AND-DISTILL META-PROMPT OPTIMIZER

A further crucial method for enhancing the efficiency of the simulation is to reduce the number of LLM requests issued by agents and the quantity of tokens consumed by said agents. A conventional approach is to reuse the generated result of a single LLM request across multiple agents. However, this approach presents two significant drawbacks: 1. In fine-grained urban LLM agent simulations, each agent possesses its own dynamic properties. Consequently, the reuse scheme compromises the independence of agents, which is antithetical to the objective of conducting urban simulations through large-scale LLM agents. Furthermore, for agents with dynamic properties, it is inherently impossible to share the result of a single LLM request, as shown in Fig.2(a).

To address this issue, we propose the Group-and-Distill Meta-Prompt Optimizer (depicted in Fig.2), which employs group information in lieu of the static attributes of the agent. This approach aggregates requests from multiple agents at runtime and realizes prompt by sharing group information and

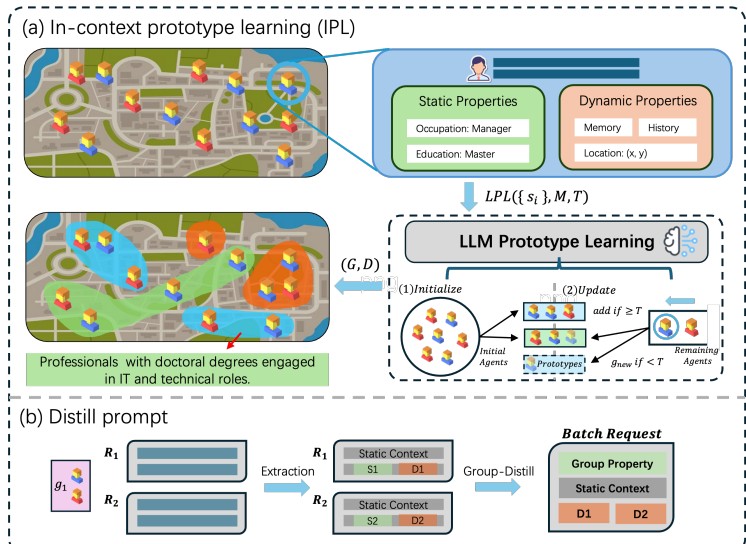

Figure 2: Overview of Group-and-Distill Meta-Prompt Optimizer.

context information while preserving the agent's dynamic properties. The optimizer is comprised of two distinct components. In-context prototype learning (IPL) and distill meta-prompt.

The inputs and outputs of IPL are defined as follow:

$$IPL(\{s_i\}, M, T) \rightarrow \mathbf{G}, \mathbf{D} \tag{2}$$

in which, $\{s_i\}$ is the collection of agent's static properties; $M$ controls the number of agents in initial prototype learning; $T$ is the threshold for decision making; $\mathbf{G}$ is the collection of agent groups; $\mathbf{D}$ is the descriptions for each group of agents.

Input the static properties of a set of agents, IPL first groups the first $M$ agents, providing both group results and the corresponding description information. Subsequently, IPL classifies the remaining agents by transmitting the static properties of the agent to LLM, which analyzes the likelihood of the agent belonging to each group based on the group description and provides the quantization result. By comparing the quantization result with $T$, when the result is greater than $T$, IPL assigns the agent to the specified group. Otherwise, it constructs a new group and describes the characteristics of the group. In comparison to conventional prototype learning methods that operate within a fixed parameter space, IPL exhibits enhanced generalization capabilities and a particular aptitude for leveraging semantic-level knowledge in the prototyping process. The prototype information obtained by IPL is employed to efficiently summarize the static attribute characteristics of the set of agents within the specified group.

The distill meta-prompts obtained through a systematic examination of the original prompts and the CoT approach is employed to generate the prompts (details can be found in Fig.A1). To facilitate the generation procedure, we have proposed a raw prompt design diagram, which divides the prompt into three sections: the function section, the variable section, and the input section. The generation process, which is initiated with a given raw prompt, comprises four steps: summarization, context extraction, information sharing, and rewriting of the raw prompt into the distill meta-prompt. In the operational phase, the requests from the agents in a group are aggregated into a single Distill request, which has the effect of reducing the number of LLM requests and the consumption of tokens.

The proposed prompt optimizer enables further enhancement of simulation efficiency and reduction of simulation cost while maintaining agent dynamic properties.

## 4.3 Web Portal

A web portal has been designed for the utilisation of OpenCity, encompassing the frontend, backend, and simulation system. This enables users to rapidly configure simulation conditions and visualise

simulation results, as well as facilitating the storage of simulation data and urban infrastructure information within a database. The fundamental concept underlying the design of this portal is user-friendliness, particularly given the inherently interdisciplinary nature of urban research. We have developed a rapid, code-free configuration approach tailored to the needs of researchers, thereby facilitating the seamless engagement of experts from diverse fields with our simulation platform.

**User-friendliness:** In order to enhance the usability of the OpenCity platform, the Web Portal has been augmented with the incorporation of the LLM agent blueprint construction function. Users are able to drag and drop each basic function module in order to construct complex logic for LLM agents. In order to meet a variety of needs, the blueprint function is based on the established LLM agent development frameworks, such as Langchain Pandya & Holia (2023) and AutoGPT Yang et al. (2023), and incorporates several fundamental modules oriented towards urban simulation, including environmental and traffic sensing. The blueprint offers an efficient and agile development solution for interdisciplinary researchers, facilitating the rapid iteration of simulation methods and theories.

**Basic workflow:** The primary process of urban LLM agent simulation on this web portal is comprised of three distinct phases: citizen profile configuration, deployment and simulation, and results presentation. The configuration of the citizen profile is facilitated by the provision of a console hub, which enables users to efficiently and transparently administer the simulation tasks they have created on the platform, along with the agents within those simulations. The user is able to bind the execution logic designed in the blueprint to different agents and to configure their profiles with great rapidity via the web interface. This may entail selecting a city, selecting an existing profile, or filling out a profile manually. Once the configuration process is complete, users can deploy and initiate simulations on the platform with a single click, leveraging the backend system and simulation system. The web portal also offers a monitor page, which enables users to observe the real-time outcomes of ongoing simulations and assess the performance of their agents. Finally, after the simulation has concluded, users can access the portal to view macroscopic statistical results in a visual format, such as Origin-Destination (OD) maps. An exemplar of the proposed web portal in operation can be found in Figure A2.

## 5 BENCHMARK

### 5.1 DATASET AND SETUP

**Dataset** We collect urban mobility data in 6 major cities around world: Beijing, New York, San Francisco, London, Paris, and Sydney. The data sources vary. Beijing's data comes from a related work Shao et al. (2024), which collected from social network platform. New York and San Francisco source from Safegraph for aggregated population flow data. And the other three cities are from Foursquare which consist of thousands of check-ins data. To make better use of these data, we have done some preprocess method, such as trajectory filter, home extraction and profile sampling. More details can be seen in Appendix A.

**Architecture of LLM Agent** The main agent used in OpenCity platform to simulate the urban dynamic is the generative agent Park et al. (2023). Generative agents use a framework that involves perception, planning, and reflection. A generative agent first creates a daily plan to ensure the trajectory is reasonable. When the agent arrives at a POI, it makes decisions based on current perceptions and memory. After taking action, the agent records the action and the POI into its memory stream. Once the memory stream reaches a threshold, the agent reflects. The results show that the generative agent to function well in the OpenCity platform.

We also have rule-based agent for comparison, such as the famous Explore and Preferential-Return (EPR) model Song et al. (2010). This work make agent choose to explore a new location or return to the visited location. Decisions are related to some parameters to compute the probability. In this paper, we set the parameters as follows: exploration rate $\rho = 0.6$, exploration-return trade-off parameter $\gamma = 0.21$, waiting time distribution parameters $\tau = 17, \beta = 0.8$.

### 5.2 ACCELERATION PERFORMANCE

This section presents an evaluation of the performance of the OpenCity platform in conjunction with the Generative Agent (Tested on Huawei ECS Cloud Server - Intel(R) Xeon(R) Platinum 8378C

CPU @ 2.80GHz with 64 cores and 256 GB RAM). The performance of the platform was evaluated in six major cities with 10,000 agents. The results are presented in Table.1, where the following variables are defined: $Speedup$ denotes the improvements in simulation time, $Rr$ denotes the LLM request number reduction rate, and $Tr$ denotes the token number reduction rate.

The results demonstrate that OpenCity exhibits substantial acceleration in all test cities, with an average runtime of 0.058s per LLM agent and an average speedup of 635.3x in simulation time. Furthermore, the proposed acceleration scheme is capable of markedly reducing the number of LLM requests and token consumption, with an average reduction of 73.7% and 45.5%, respectively.

To assess the scalability of OpenCity, we conducted a series of simulations to evaluate its acceleration performance under varying orders of magnitude of agents. The results of this analysis are presented in Fig.3, in which the baseline represents the simulation time without optimization. The results demonstrate that OpenCity's acceleration capability is scalable, with a notable enhancement in acceleration effect when the number of agents is increased from 10 to 10,000. This is due to the fact that as the number of agents increases, the number of groups obtained based on IPL also gradually increases. This, in turn, allows the advantages of the LLM request scheduler to be fully realised, thereby ensuring a better utilisation of system resources.

| Cities | Time | $Speedup$ | $Rr$ | $Tr$ |
|---|---|---|---|---|
| Beijing | 0.07s | 521.7 | 73.2% | 38.7% |
| New York | 0.06s | 624.7 | 67.3% | 37.6% |
| San Francisco | 0.07s | 588.6 | 80.3% | 51.3% |
| London | 0.04s | 792.5 | 74.6% | 49.9% |
| Paris | 0.06s | 640.0 | 76.3% | 48.6% |
| Sydney | 0.05s | 644.0 | 70.7% | 46.6% |
| **Average** | **0.058s** | **635.3** | **73.7%** | **45.5%** |

Table 1: Acceleration experiment results

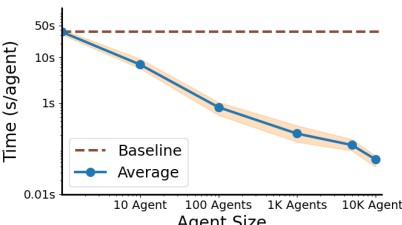

Figure 3: Scalability experiments

Furthermore, faithfulness experiments are conducted to demonstrate that the Group-and-Distill optimizer can effectively preserve the distinctive personality traits of the agents. The testbed for this evaluation is location choice generation, which requires the combination of agent properties to select the next location to visit. A comparison was conducted between the performance of four distinct methods, including raw prompting (without any modification), batch prompting Cheng et al. (2023), archetype prompting Chopra et al. (2024), and the proposed method. One hundred agents were randomly selected and location selection was performed 100 times for each agent with the same context. The effectiveness of the method was evaluated by counting the distribution of selections ($JSD$) as well as the top-1 hit rate ($T1$). The results are shown in Table.2, where Inherent denotes the bias present in LLM itself (raw prompt method).

| Model and Cities | | Inherent $JSD$ | $T1$ | Batch prompting $JSD$ | $T1$ | Archetype prompting $JSD$ | $T1$ | Ours $JSD$ | $T1$ |
|---|---|---|---|---|---|---|---|---|---|
| 4o-mini | BJ | $0.04 \pm 0.02$ | 90% | $0.11 \pm 0.05$ | 76% | $0.89 \pm 0.04$ | 8% | $0.13 \pm 0.02$ | 74% |
| | NY | $0.02 \pm 0.01$ | 92% | $0.07 \pm 0.03$ | 81% | $0.84 \pm 0.11$ | 13% | $0.06 \pm 0.04$ | 86% |
| | SF | $0.03 \pm 0.02$ | 88% | $0.09 \pm 0.04$ | 77% | $0.91 \pm 0.03$ | 11% | $0.10 \pm 0.03$ | 85% |
| | Lo | $0.06 \pm 0.04$ | 89% | $0.12 \pm 0.07$ | 79% | $0.86 \pm 0.06$ | 9% | $0.12 \pm 0.04$ | 78% |
| | Pa | $0.05 \pm 0.02$ | 86% | $0.17 \pm 0.11$ | 69% | $0.94 \pm 0.03$ | 4% | $0.14 \pm 0.04$ | 71% |
| | Sy | $0.04 \pm 0.03$ | 85% | $0.08 \pm 0.03$ | 75% | $0.88 \pm 0.05$ | 5% | $0.07 \pm 0.04$ | 75% |
| GPT-4o | NY | $0.003 \pm 0.002$ | 98% | $0.012 \pm 0.007$ | 94% | $0.89 \pm 0.09$ | 10% | $0.009 \pm 0.004$ | 97% |
| | Pa | $0.004 \pm 0.002$ | 99% | $0.021 \pm 0.009$ | 93% | $0.91 \pm 0.04$ | 7% | $0.010 \pm 0.006$ | 96% |

Table 2: Faithfulness experiment results

As evidenced by the results, our method demonstrates the capacity to maintain a comparable level of consistency to that observed in the batch prompting method, while exhibiting a reduction in volatility and token consumption. However, the archetype prompting method performs poorly in this evaluation, which further demonstrates the inability of the reuse-based method to accommodate the dynamic properties of agents. Furthermore, given the considerable discrepancies observed in the raw prompting method when evaluated using the GPT-4o-mini model, an additional assessment was conducted on two cities, New York and Paris, utilising the GPT-4o model. The findings indicate that our method is capable of approximating the execution of the raw prompting method to a

significant degree. Additionally, the results indicate that there are notable discrepancies between different models in terms of environmental comprehension and the capacity to process lengthy textual content. The consistency of LLM outcomes merits further examination.

In general, OpenCity is capable of markedly enhancing the efficiency of large-scale urban LLM agent simulations while concurrently preserving the distinctive characteristics of the agents themselves. This enables the cost of simulating populations exceeding 10,000 to be maintained at the hourly level.

## 5.3 Reproducing Urban Dynamics

The significant increase in simulation efficiency enables us to benchmark LLM agent's ability to reproduce large-scale urban dynamics for the first time. We use comprehensive metrics in three-levels to evaluate the simulation performance, from individual- to group level, and also from physical domain to social domain. At the individual level, we calculate the radius of gyration Gonzalez et al. (2008) for each user. At the group level, we use the original-destination matrix Jiang et al. (2016). As for the social domain, we focus on the income segregation index Moro et al. (2021). To evaluation the simulation performance, we compute the MSE for these three metrics, which are denoted as $R_{MSE}$, $OD_{MSE}$ and $S_{MSE}$. More details can be referred to Appendix B.

In this section, we analyze the performance of the Generative Agent and EPR Agent in reproducing urban dynamics. We test both agents in 6 major cities using 1,000 agents. The results are shown in Table 3. The results indicate that both the Generative Agent and EPR Agent successfully reproduce urban dynamics with low MSE values. Additionally, the LLM Agent performs as well as or better than the classical rule-based EPR Agent, highlighting the advantage of LLM's semantic understanding ability in urban simulations.

| Cities | GenerativeAgent | | | EPR | | |
|---|---|---|---|---|---|---|
| | $R_{MSE}$ | $OD_{MSE}$ | $S_{MSE}$ | $R_{MSE}$ | $OD_{MSE}$ | $S_{MSE}$ |
| Beijing | 19.5 | 3.88e-4 | 0.0312 | 29.8 | 4.26e-4 | 0.0630 |
| New York | - | 5.95e-4 | 0.3521 | - | 3.70e-4 | 0.2319 |
| San Francisco | - | 23.6e-4 | 0.1535 | - | 14.0e-4 | 0.0352 |
| Paris | 2.48 | 7.58e-4 | 0.1255 | 4.04 | 6.25e-4 | 0.1240 |
| London | 6.24 | 5.22e-4 | 0.1258 | 25.7 | 7.41e-4 | 0.1501 |
| Sydney | 15.1 | 4.71e-4 | 0.1118 | 54.2 | 7.63e-4 | 0.1265 |

Table 3: Urban dynamics reproduction results

# 6 Case Study: Experienced Urban Segregation

With the ability to simulate large-scale urban LLM agents, we can conduct counterfactual experiments to explore outcomes under different policies and design optimal strategies for the future. Conventional rule-based models do not support this capability, as they are designed to simulate real-world scenarios. Experienced urban segregation is a widely discussed issue with significant impacts on social dynamics and the economy. It arises from both demographic differences in residential neighborhoods and the mobility patterns of urban residents Moro et al. (2021). This section provides a case study: a counterfactual simulation is conducted in New York and San Francisco, to observe how the simulation results change in different configurations, and try to summarize the results with the LLM agents themselves.

Specifically, we construct the counterfactual scenario by evenly distributing LLM agents with different income levels across the city, that means we almost eliminate the residential segregation. The results of the income segregation statistics with CBGs as the statistical granularity are shown in Fig.4, where 'Original' samples the segregation results from the real census data, and 'Even' is the result after uniform distribution of agents with different incomes.

From the results, it can be seen that the segregation of the two cities changed significantly after the different income groups were evenly distributed in the cities. In New York City, the mean segregation index decreases from 0.845 to 0.172, and in San Francisco, the mean segregation index

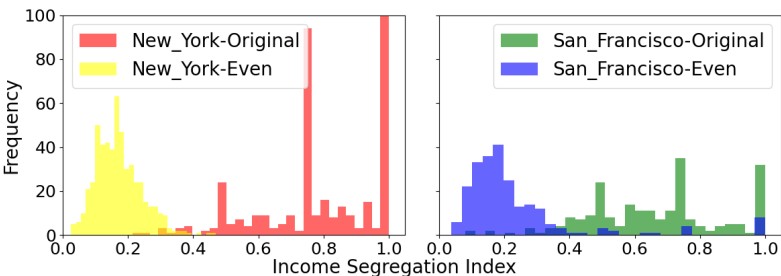

Figure 4: The distribution of income segregation index for counterfactual experiment.

decreases from 0.665 to 0.232. As a result, we believe that differences between regions are the main cause of segregation as opposed to segregation by choice of action. To extend, we can know that with policies that promote more even income distribution among neighborhoods, urban segregation and social inequality can be improved.

| Human Input | What places did you go today and why? | How many people did you see today? How were they? |
|---|---|---|
| Agent Output | About the places I visited today and the reasons:
1.Sleep at home (00:00 - 10:00): Resting to start the day.
2.Brunch (10:00 - 11:32): Enjoying a leisurely meal.
3.Outdoor Activity (12:06 - 14:45): Engaging in physical activity
4.Curry Boyz (15:43 - 17:10): Likely for a meal, perhaps trying Indian cuisine.
5.Walgreens (17:53 - 20:38): To pick up personal care items or medication.
6.Blush! Wine Bar (21:23 - 23:02, and stayed until late): Relaxing, enjoying wine, and possibly socializing.
Each place was chosen for a mix of relaxation, exercise, dining, and socializing. | The people I met today:
1. When I went to Curry Boyz, I met [547] and [1032], [547] is an IT engineer, [1032] is a teacher, [547] has a high level of consumption, [1032] has a medium level of consumption.
2. When I went to Blush! Wine Bar, I met 1032 and 779, [779] is a worker, [779]'s consumption level is slightly low.
The above is about the people I met today and what they were up to. |

Figure 5: A detail case of interpreting simulation results through communication.

Furthermore, we use natural language to communicate with those involved agents to gain deeper insights about urban segregation. One detailed case is shown in Fig.5. When we ask an agent about its daily journey, it can accurately provide the time and locations it visited. This is because the agent caches runtime information and uses the LLM's ability to understand semantic details. When asked about the people it met, the agent lists everyone it encountered at different locations and provides their information. This is due to vectorized storage of the agent's simulation results and the LLM's ability to retrieve that information. Collecting and observing fine-grained statistical information through conversations with agents and even through LLM improves both the interpretability of the simulation and our understanding of the simulation goals.

## 7 CONCLUSION

In this work, we introduced OpenCity, a scalable platform designed to address the computational and communication challenges inherent to the deployment of large-scale LLM-based urban agents in city simulations. By incorporating an LLM request scheduler and a novel "group-and-distill" prompt optimization strategy, we achieved a notable 600-fold increase in the efficiency of agent simulations, with a substantial reduction in both LLM requests and token usage. The OpenCity platform was evaluated through experiments conducted on six global cities. The results demonstrated the platform's capability to simulate the daily activities of 10,000 agents at an hourly level, while also establishing a benchmark for generative agent performance in urban contexts. The platform's ability to compare simulated behaviors with real-world data highlights its potential for real-world urban-scale applications, offering a robust tool for urban planners and researchers to explore and understand complex societal phenomena.

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

## A URBAN MOBILITY DATASET

As shown in Table A1, we collect urban mobility data of 6 major cities around the world. The data sources vary. In Beijing, the data is from a related work Shao et al. (2024), which gathered through a social network platform and tracking users' mobility trajectories. Additionally, users' profiles, such as income level, gender, occupation, education level and age, are collected through digital surveys. In New York and San Francisco, the data comes from Safegraph, which provides aggregated population flow among Points of Interest(POIs) and Census Block Groups(CBGs). The other three cities—London, Paris and Sydney—use data are from Foursquare. Foursquare data consist of thousands of check-ins data of users and the corresponding venue position.

To make better use of the datasets, we apply several preprocessing methods. We firstly arrange the trajectory points in time sequence, and divide the trajectory into units of one day. Then we filter out trajectories with fewer than 4 points in a day, as they do not fully capture users' mobility patterns. For home extraction, we identify the most frequently visited location of the useres the home. Since only the Tencent dataset includes user profiles, we make profile sampling for users of each city based on local census data for the other two datasets. In the end, our dataset is optimized for easy use in urban mobility simulations.

| Source | City | Users | Trajectory Points | Duration |
|--------|------|-------|-------------------|----------|
| Shao et al. (2024) | Beijing | 100000 | 297363263 | Oct. 2019 - Dec. 2019 |
| Safegraph | New york
San Francisco | Aggregated
Aggregated | 760493
316732 | May 2023 - July 2023 |
| Foursquare | London
Paris
Sydney | 9409
5809
1720 | 173268
85679
54170 | Apr. 2012 - Sept. 2013 |

Table A1: Basic information about the dataset

## B  URBAN DYNAMIC METRICS

We use comprehensive metrics in three-levels to evaluate the simulation performance, from individual-level to group level, and also from physical domain to social domain. These metrics allows us to gain a full understanding of mobility patterns and their implications, and can also help us evaluate the performance of the simulation by analysing the generated trajectory.

At the individual level, we calculate radius of gyration $r_g$ Gonzalez et al. (2008) for each user, which is a measure of the spatial extent of their movements. The radius of gyration is defined as follows:

$$r_g^\alpha = \sqrt{\frac{1}{N^\alpha} \sum_{i=1}^{N^\alpha} (\vec{r}_i^\alpha - \vec{r}_{cm}^\alpha)^2} \qquad (3)$$

where $\vec{r}_i^\alpha$ represents the $i = 1, 2, ..., N$ positions recorded by user $\alpha$, and $r_{cm}^\alpha = 1/N^\alpha \sum_{i=1}^{N^\alpha} (\vec{r}_i^\alpha)$ is the center of mass of the trajectory. The radius of gyration provides an indication of the size of a user's activity range. To assess the accuracy of our simulation data against real-world data for a specific user, we calculate $R_{MSE}$, the Mean Squared Error(MSE) of the radius of gyration.

To analyze movement patterns and other aggregated features, we define block areas as spatial units within the city. For cities with Safegraph data, we use existing Census Block Group (CBG) areas. For other cities, we divide the map area into evenly spaced grids, with each grid cell representing a block area.

At the group level, we count the inflow and outflow of agents between block areas, calculate the Origin-Destination (OD) matrix Jiang et al. (2016), and normalize it. To compare real data with simulation data, we calculate the MSE of the normalized OD matrix, denoted as $OD_{MSE}$. A smaller $OD_{MSE}$ value indicates greater similarity between the OD matrices, meaning the movement characteristics of the simulated data closely match the real data.

At the social domain, we calculate the income segregation index Moro et al. (2021) for each block area. The income segregation of a place $\alpha$ is defined as $S_\alpha = \frac{5}{8} \sum_q |\tau_{q\alpha} - \frac{1}{5}|$, where $\tau_{q\alpha}$ is is the proportion of visitors in each income quintile for place $\alpha$. The $S_\alpha$ ranges from 0 to 1. A high $S_\alpha$ indicates that the place $\alpha$ is predominantly visited by a single income group, suggesting a high level of income segregation. We denote $S_{MSE}$ as the MSE between the real data and simulation data.

## C  IMAGE SUPPLEMENTS

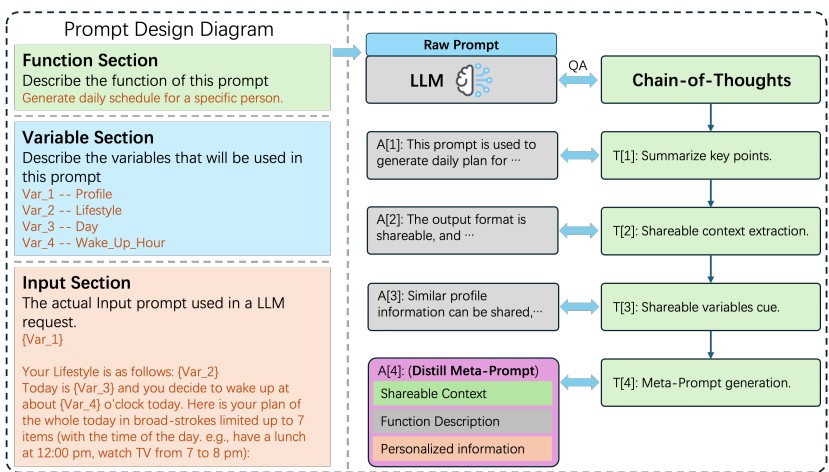

Figure A1: Distill meta-prompt generation through CoT inference.

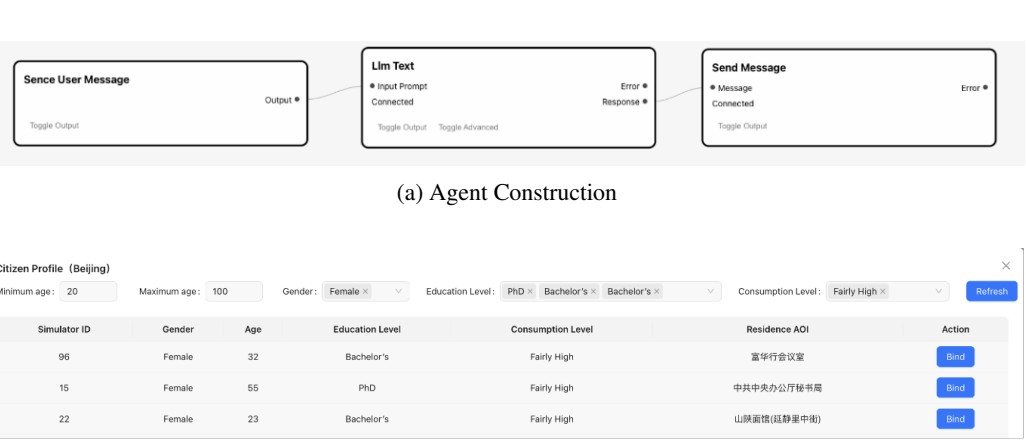

(a) Agent Construction

(b) Profile Configuration

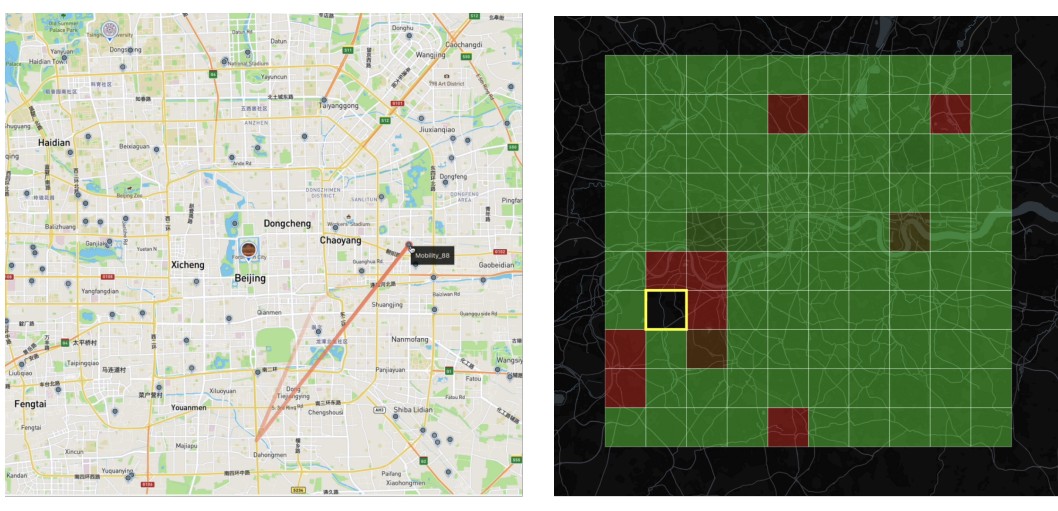

(c) Simulation Visualization    (d) Result Visualization

Figure A2: Overview of OpenCity web portal.

