# OpenReview forum: "OpenCity: A Scalable Platform to Simulate Urban Activities with Massive LLM Agents"
_ICLR.cc/2025/Conference — Submitted to ICLR 2025_

### Official Review · Reviewer_eoM2 · 2024-10-21

**Soundness:** 2
**Presentation:** 2
**Contribution:** 2
**Rating:** 5
**Confidence:** 4

**Summary:**

ABM and LLM is leveraged to develop one platform for open city modeling and planning. It is a nice simulation platform and the paper provides application scenarios. Concretely,

1. This paper combines agent-based models with large language models to develop the OpenCity platform for simulating urban activities. It reduces simulation computational costs through IO multiplexing and the "group-and-distill" prompt optimization strategy.
2. Through experiments conducted in six major cities worldwide, OpenCity demonstrates a 635 plus increase in average simulation speed per agent, along with a 70% decrease in LLM requests and a 50% reduction in token usage. The time savings are mainly concentrated in the LLM response wait time and the CPU multiplexing process.
3. The OpenCity platform proposed in this paper achieves the first benchmark testing for LLM agent-based urban activity simulation research.

**Strengths:**

The paper give a detailed introduction of the novel methods and the real outcome.

1. Originality: The paper presents a novel approach by integrating Large Language Models with agent-based modeling to simulate urban activities. There has been limited research on combining LLMs with agent-based models, and even less so in the context of large-scale urban activity simulations. By using IO multiplexing and the "group-and-distill" prompt optimization strategy to reduce the computational cost of simulations, this paper has made the application of LLMs in large-scale urban activity simulations possible.
2. Quality: The research is well organized with a clear methodology and experiments conducted in real cities data. The results show notable improvements in both simulation efficiency and accuracy, confirming the effectiveness of the proposed platform.
3. Clarity: The paper is written in a clear and concise manner; it is easy to understand through the explanation of figures
4. Significance: This paper establishes a benchmark for LLM agent-based urban activity simulation research. This paper also provides a scalable framework for simulating urban dynamics.

**Weaknesses:**

There is a lack of theoretical contribution, overall, rather it is an application tool development with leveraging well established tools. It may not fit ICLR the best though not out of scope at all. Further,

1. Some parts of the main body text are not rigorous enough. For example, Equation 1 is missing a parenthesis, and the IPL method is mistakenly labeled as the LPL method in Figure 2.
2. This research has high requirements for data quality. Additionally, despite significantly improving computational efficiency and reducing costs, the platform may still require substantial computational resources.
3. When simulating cities in different countries, the dynamic properties to be considered should not be entirely the same, and some of the assumed static properties may also change during the simulation process.

**Questions:**

1.	It seems that the formula in Equation 1 misses a parenthesis?  And it can also be split into two separate equations.
2.	In Section 3.2, the content of Figure 1(b) is introduced, however, the Figure 1(a) is discussed in Section 4.1. Is this logical? It is suggested to swap the subtitles of the figures for better coherence.
3.	This paper categorized the CPU task as ”local IO”, offload it to available cores for computation through a multi-core parallel scheme, and then return the result to the designated agent upon completion of the computation.
4.	Is there an increase in CPU task scheduling time? How does it compare to the time saved in saving#3?
5.	The paper states that "A conventional approach is to reuse the generated result of a single LLM request across multiple agents, and this approach presents two significant drawbacks." However, it only describes one drawback.
6.	The paper introduces Figure 2(a) first, which prompts readers to check the content of Figure 2(a). However, they will encounter a series of unexplained equations, which could cause confusion. The paper cites Figure 2 only in the subsequent description, and the explanation of Figure 2(a) is found only in the explanation of the methodology of Figure 2 in the main text. Is it possible to optimize this part of the description?
7.	From the subsequent description in the main text, it is understood that Figure 2(a) is introducing the IPL method, but in the figure is labeled as the LPL method.
8.	How is the threshold T in the IPL method of this paper determined?
9.	Are the baseline in Figure 3 increased simulation response time due to the load as the number of agents increases?
10.	In Table 3, there is no explanation for the missing RMSE results for New York and San Francisco. According to Appendix A, the data for these two cities comes from Safegraph, and the number of users is aggregated. So how is the GROUP-AND-DISTILL performed in these two cities?

---

> ### Author Response · Authors · 2024-11-24
> **Response to Reviewer eoM2 (Part1)**
>
> Thank you for your valuable suggestions. We truly appreciate the time and effort you put into reviewing our work. Your feedback has been incredibly helpful, and we will incorporate your suggestions to improve the clarity and depth of the paper. We provide a point-by-point response below:
>
> ### Weaknesses
> **W1: Some parts of the main body text are not rigorous enough. For example, Equation 1 is missing a parenthesis, and the IPL method is mistakenly labeled as the LPL method in Figure 2.**
>
> **Response:** We appreciate your attention to detail. We acknowledge the issues you pointed out: the missing parenthesis in Equation 1 and the incorrect labeling of the IPL method as LPL in Figure 2. We apologize for these oversights and will correct them in the revised version of the paper. These corrections will help improve the clarity and rigor of the presentation.
>
> **W2: This research has high requirements for data quality. Additionally, despite significantly improving computational efficiency and reducing costs, the platform may still require substantial computational resources.**
>
> **Response:** Thank you for your thoughtful feedback.
>
> + Regarding the data quality issue: The core focus of this work is to enhance the simulation efficiency of large-scale LLM agents. While data quality does influence the results of LLM agent simulations, it does not affect the main contribution of this paper, which is improving the efficiency of agent simulations. Additionally, the data used in this work comes from open-source datasets such as Census and Foursquare. The validity of the original data is not a primary focus of this study; rather, it serves as a foundation for large-scale LLM agent simulations. The experiments conducted aim to demonstrate the effectiveness of the proposed method in improving simulation efficiency.
> + On the issue of computational resources: Thank you for your comments. Our paper does not emphasize the deployment scenarios of the simulation framework (only briefly mentioned in the experimental section). In practice, OpenCity is designed for user-side simulation needs, where LLM calls are made through remote service requests (e.g., OpenAI). In this context, the primary computational consumption is related to network I/O and CPU resources, as LLM inference is handled by the service provider. Therefore, using this method for large-scale agent simulations imposes a low computational resource requirement on the user’s side.
>
> **W3: When simulating cities in different countries, the dynamic properties to be considered should not be entirely the same, and some of the assumed static properties may also change during the simulation process.**
>
> **Response:** Regarding the distinction between dynamic and static attributes: In this work, we categorize the attributes of LLM agents into static and dynamic properties. However, this categorization is inherently tied to the problem at hand and the granularity of change. For example, in the mobility simulation task presented in the experimental section, where the goal is to simulate human activity trajectories throughout a day, we assume that attributes like gender and income level are static, while dynamic attributes include elements like memory. On the other hand, in simulations with longer time spans, static attributes may evolve into dynamic ones (e.g., age), and a more reasonable approach may be to update the agent’s “dynamic attributes” at fixed time intervals based on the granularity of updates. The goal of OpenCity is to establish an open-source enabling framework that facilitates more fine-grained modeling for researchers across various disciplines.
>
> ### Questions
> **Q1: It seems that the formula in Equation 1 misses a parenthesis? And it can also be split into two separate equations.**
>
> **Response:** We acknowledge the issue with Equation 1, where a parenthesis is missing. We apologize for this oversight and will correct it in the revised version. Additionally, we appreciate your suggestion to split the equation into two separate parts. We agree that doing so could enhance clarity, and we will revise the equation accordingly to improve its readability and understanding.
>
> **Q2: In Section 3.2, the content of Figure 1(b) is introduced, however, the Figure 1(a) is discussed in Section 4.1. Is this logical? It is suggested to swap the subtitles of the figures for better coherence.**
>
> **Response:** We acknowledge the inconsistency you pointed out regarding the order of Figure 1(a) and Figure 1(b) in the paper. We agree that this may cause confusion in the logical flow. Based on your suggestion, we will swap the subtitles of the figures to ensure better coherence and alignment with the sections in which they are discussed.

---

> ### Author Response · Authors · 2024-11-24
> **Response to Reviewer eoM2 (Part2)**
>
> **Q3: This paper categorized the CPU task as ”local IO”, offload it to available cores for computation through a multi-core parallel scheme, and then return the result to the designated agent upon completion of the computation. Is there an increase in CPU task scheduling time? How does it compare to the time saved in saving#3?**
>
> **Response:** In this work, the CPU task scheduling strategy is relatively simple: in a system with N cores, we designate one core as the IO core, which is responsible for managing LLM remote service requests, while the remaining N-1 cores handle the CPU computation tasks. Additionally, when LLM agents generate CPU tasks during execution, OpenCity employs a greedy approach to assign the tasks to the computational cores, prioritizing the cores with the least amount of work or those that are idle.
>
> + **Regarding task scheduling and computation time:** Task scheduling time is closely tied to the performance of the operating system but typically remains at the **microsecond level**. Overall, task scheduling time is relatively short and can generally be considered a fixed overhead. In contrast, the computation time required for tasks varies dynamically with task complexity (In OpenCity, when using the gravity model for location selection with a range of 1000 meters, the average computation time per calculation is approximately **0.13 seconds**.) and may even extend to several seconds.
>
> **Q4: The paper states that "A conventional approach is to reuse the generated result of a single LLM request across multiple agents, and this approach presents two significant drawbacks." However, it only describes one drawback.**
>
> **Response:** Thank you very much for your response and for pointing out this issue. We truly appreciate your careful review. Regarding your question, in addition to the limitation already mentioned in the paper, another drawback of the LLM request reuse method is that it cannot adapt to dynamic environments. For instance, when dealing with location selection problems, even if all LLM agents have identical attributes, the availability of options can vary. In such cases, this reuse method inherently cannot serve these types of problems.
>
> **Q5: The paper introduces Figure 2(a) first, which prompts readers to check the content of Figure 2(a). However, they will encounter a series of unexplained equations, which could cause confusion. The paper cites Figure 2 only in the subsequent description, and the explanation of Figure 2(a) is found only in the explanation of the methodology of Figure 2 in the main text. Is it possible to optimize this part of the description?**
>
> **Response:** We appreciate your suggestion regarding the explanation of Figure 2(a). We agree that introducing Figure 2(a) before providing a clear explanation of the equations can lead to confusion. To address this, we will revise the paper to ensure that Figure 2(a) is accompanied by a more immediate and detailed explanation. This will help readers better understand the figure and its equations in the context of the methodology, avoiding any confusion when they first encounter it.
>
> **Q6: From the subsequent description in the main text, it is understood that Figure 2(a) is introducing the IPL method, but in the figure is labeled as the LPL method.**
>
> **Response:** We acknowledge the labeling error in Figure 2(a), where it was mistakenly labeled as the LPL method instead of the correct IPL method. We apologize for this oversight and will correct the figure label to accurately reflect the IPL method in the revised version of the paper.
>
> **Q7: How is the threshold T in the IPL method of this paper determined?**
>
> **Response:** In this work, T is a hyperparameter in the IPL method, which controls the granularity of the agent grouping. Specifically, as T increases, the number of agent groups increases, and conversely, as T decreases, the groups become fewer. At this stage, we have not conducted an in-depth study of this hyperparameter, but as you rightly pointed out, T directly influences the grouping results in the IPL method. We appreciate your suggestion, and in the revised version, we plan to include a more detailed analysis of this hyperparameter, exploring its impact on agent grouping and its role in the method’s performance.

---

> ### Author Response · Authors · 2024-11-24
> **Response to Reviewer eoM2 (Part3)**
>
> **Q8: Are the baseline in Figure 3 increased simulation response time due to the load as the number of agents increases?**
>
> **Response:** First, we would like to apologize for the confusion. Due to the long runtime of the baseline method, it has been challenging to scale it to larger numbers of agents (1000+). We conducted experiments with a baseline method for 1000 agents, and the reported baseline time is actually the average execution time for 1000 agents (i.e., total execution time divided by 1000). During the experiment, we observed that the fluctuation in the average execution time for 1000 agents is consistent with the fluctuation observed when running the same agent multiple times. This is primarily due to the inherent variability in remote service calls. Thank you again for your valuable feedback. We will include a clarification of this data in the revised version of the paper.
>
> **Q9: In Table 3, there is no explanation for the missing RMSE results for New York and San Francisco. According to Appendix A, the data for these two cities comes from Safegraph, and the number of users is aggregated. So how is the GROUP-AND-DISTILL performed in these two cities?**
>
> **Response:** About the missing $ R_{MSE} $ of New York and San Francisco: The absence of $ R_{MSE} $ for New York and San Francisco is due to the nature of the dataset in these cities, which aggregates user data without individual trajectories. Since $ R_{MSE} $ measures the difference in radius of gyration between real and simulated users, it is not applicable in these cases. We will clarify this in the manuscript. Regarding how the Group-and-Distill method affects the LLM agent simulation in these two cities, its impact is primarily reflected in the improvement of simulation performance. Specifically, by aggregating multiple LLM agent requests into a single request, the method reduces the number of LLM requests and the associated token consumption. At the same time, it preserves the personalized attributes of the LLM agents to maintain their original performance. The effectiveness of this approach in retaining agent individuality and enhancing performance can be observed in Section 5.2.

---

> ### Comment · Reviewer_eoM2 · 2024-11-25
>
> Based on authors' response, I am happy to keep my rating. Hope authors can address the required corrections and clarifications suggested by reviewers in the revised manuscript.

---

### Official Review · Reviewer_WDiW · 2024-10-26

**Soundness:** 2
**Presentation:** 3
**Contribution:** 2
**Rating:** 3
**Confidence:** 4

**Summary:**

The paper is considering the problem of agent-based modeling of environments such as cities. Such models had been previously used with the agents performing relatively simple behaviors. While LLMs open new opportunities for controlling the behaviors of the individual agents, their computational cost presents a significant scaling problem.

The paper describes an architecture that enables the parallelization of the agents, to allow the modeling the daily activities of a city with 10,000 agents. The architecture appears to be based on an efficient polling model of the LLM, as well as the development of a prompting model called "group-and-distill". The application of these models show a more than 600-fold increase.

**Strengths:**

* The overall goals of the paper, of capitalizing on the abilities of LLMs to achieve a better ABM model of cities, as well as addressing the scaling problems, are laudable.

**Weaknesses:**

* Achieving a more than 600 fold speed increase in terms of an improved process scheduler and I/O scheduler can be probably seen as "debugging", rather than research result, and very likely has nothing to do with the LLM.
* It seems that the very considerable computational effort of an LLM can only achieve an approximate parity with the much cheaper rule based efforts. This is understandable, as description of the behavior of the agents described in the paper follows the same position based rules that the ABM models historically use. As there is no consideration of language or other type of reasoning, the paper does not make it clear what type of benefits one would expect from LLMs.
* The only part of the paper that has a connection to the topic of this conference is the way in which the "group-and-distill" model is proposed to achieve the simulation of multiple agents with one prompt. However, there is very little about this technique in the paper proper, so it is difficult to form a judgement.

**Questions:**

* The paper spends comparatively less effort on explaining what kind of benefits do we expect from an LLM-based ABM. For instance, we can try to model thought processes of the humans, or their communication. Does the choice of this modeling impact the proposed techniques?

---

> ### Author Response · Authors · 2024-11-24
> **Response to Reviewer WDiW (Part1)**
>
> Thank you for your valuable feedback.
>
> ### Weaknesses
> **W1: Achieving a more than 600 fold speed increase in terms of an improved process scheduler and I/O scheduler can be probably seen as "debugging", rather than research result, and very likely has nothing to do with the LLM.**
>
> **Response:** In the context of large-scale LLM agents simulation, it involves massive remote service requests. To enhance the overall efficiency of simulation operations, the focus should not solely rest on improving the performance of LLM itself. We can also optimize the execution of tasks within the paradigm of large-scale LLM invocations through various methodologies. Specifically, for your concern:
>
> + **For 600-fold speed increase:** The 600-fold speed improvement is not solely attributed to the LLM request scheduler; it also stems from the optimization of the Group-and-Distill algorithm.
> + **For the design of the LLM request scheduler:** The proposed request scheduler is not designed for any specific LLM, but rather tailored for this particular scenario. For example, it reduces the waiting time for LLM requests during agent execution by handling concurrent requests, and it minimizes system overhead by sharing remote LLM service connections.
>
> **W2: It seems that the very considerable computational effort of an LLM can only achieve an approximate parity with the much cheaper rule based efforts. This is understandable, as description of the behavior of the agents described in the paper follows the same position based rules that the ABM models historically use. As there is no consideration of language or other type of reasoning, the paper does not make it clear what type of benefits one would expect from LLMs.**
>
> **Response:** First, in this paper, we have never claimed that LLM Agent-based approaches significantly outperform traditional rule-based methods, particularly with regard to a specific agent design approach, which in this paper refers to the Generative Agent. Besides, the effectiveness of ABM methods in understanding complex systems has been demonstrated in several studies (for example, "AgentTorch: Agent-based Modeling with Automatic Differentiation," which has been published in NeurIPS 2024: [https://neurips.cc/virtual/2023/79209](https://neurips.cc/virtual/2023/79209)). In fact, the core goal of this work is to enhance the performance of large-scale agent simulations and reduce computational costs, thus supporting ABM based on LLM agents.
>
> + **In Section 5.2**, we provide experimental evidence showing that the method proposed by OpenCity has minimal impact on the performance of the original LLM agents, which makes it feasible to support simulation work. In Section 5.3, while Generative Agents may not offer significant advantages over the EPR method, this is directly related to the design of the agents and does not affect the core contribution of this paper.
> + **In the Case Study section**, we conduct counterfactual experiments and agent interviews, which highlight the advantages of LLM agents over traditional methods, particularly in terms of semantic understanding and reasoning capabilities.
>
> **W3: The only part of the paper that has a connection to the topic of this conference is the way in which the "group-and-distill" model is proposed to achieve the simulation of multiple agents with one prompt. However, there is very little about this technique in the paper proper, so it is difficult to form a judgement.**
>
> **Response:** Thank you for your insightful feedback. We appreciate your recognition of the relevance of the “group-and-distill” model, which is central to our approach of simulating multiple agents with a single prompt. We acknowledge that the discussion of this technique in the paper may not have been as detailed as it should be, and we understand how this could make it difficult to fully evaluate its contribution. To address this, we will expand on the “group-and-distill” model in greater detail, specifically explaining its design, implementation, and the advantages it offers in the context of large-scale agent simulation. By providing more clarity and examples, we aim to make the connection to the conference theme more explicit and allow a better understanding of how this technique enhances the simulation of multiple agents.

---

> ### Author Response · Authors · 2024-11-24
> **Response to Reviewer WDiW (Part2)**
>
> ### Questions
> **Q1: The paper spends comparatively less effort on explaining what kind of benefits do we expect from an LLM-based ABM. For instance, we can try to model thought processes of the humans, or their communication. Does the choice of this modeling impact the proposed techniques?**
>
> **Response:** The primary goal of this work is to enhance the simulation efficiency of large-scale LLM agents and reduce the computational cost, making large-scale ABM simulations based on LLM agents feasible. While human thought processes, such as cognitive functions or communication, are indeed important aspects, the focus of this work is not on the specific design of the agents themselves but rather on improving the overall simulation efficiency and scalability.
>
> The modeling of human thoughts is closely tied to the design of the agents. In our experiments, we used Generative Agents (GA), which inherently include aspects of cognitive processes, such as action intention generation and memory management. These elements are part of the GA’s design and are integral to how the agents simulate decision-making and behavior.
>
> We appreciate your suggestion, and based on your feedback, we will revise the paper to include more detailed information about the agent design. This will help readers better understand the work and its contributions in the context of large-scale LLM agent simulation.

---

> > ### Comment · Reviewer_WDiW · 2024-11-28
> > **The author response does not change my evaluation.**
> >
> > The author response does not change my evaluation.

---

### Official Review · Reviewer_bSG3 · 2024-11-01

**Soundness:** 3
**Presentation:** 2
**Contribution:** 2
**Rating:** 6
**Confidence:** 3

**Summary:**

The recent rise of Large Language Models (LLMs) has led to the development of LLM agents capable of simulating urban activities with unprecedented realism. Nevertheless, the extreme high computational cost of LLMs presents significant challenges for scaling up the simulations of LLM agents. With this motivation, this paper introduces OpenCity, a scalable simulation platform designed to efficiently simulate urban activities using a large number of LLM agents. The platform incorporates innovative techniques, including LLM request scheduler and a group-and-distill prompt optimization strategy, to reduce the computational overhead of simulating LLM agents significantly. OpenCity achieves a 600-fold speedup and reduces both LLM requests and token consumption. Extensive experiments on six global cities verify the platform's scalability and its capability to replicate real-world urban dynamics.

**Strengths:**

Strengths
1.	This paper introduces a scalable platform for urban simulations using LLM agents, which addresses a growing need for realistic human behavior modeling in urban environments.
2.	This paper shows a high quality of presentation. The paper is technically sound and the research question is clear. The optimizations, particularly the LLM request scheduler and prompt optimization strategies, demonstrate clear performance benefits. The experimental results showing a 600x speedup and significant resource savings are compelling.
3.	The paper is generally clear and well-structured. It provides a clear problem statement, introduces the proposed framework, and highlights key findings.
4.	The contribution of the paper is relevant for LLM agent. The results of this paper is interesting and significant in LLM agent simulation. The proposed OpenCity framework is relevant for urban planning and policy-making. The development of a web portal that allows researchers to configure and visualize simulations without requiring programming skills is a valuable addition, making the platform accessible to a broader audience.

**Weaknesses:**

1.	The introduction part fails to convey to the reviewers what is the motivation and novelty in this paper. In fact, the authors should add more previous work on LLM agents based simulation platform. The problem this paper addresses and the reason why this paper uses system-level LLM request scheduler and prompt-level “group-and-distill” strategy to solve the problem of scalability should be further explained. Besides, the contribution the authors listed in the introduction section is inaccurate，the authors should focus on the system-level LLM request scheduler and prompt-level “group-and-distill” strategy. Thus, I would recommend a revision for the introduction section in this paper.
2.	This paper utilizes Group-and-Distill Meta-Prompt Optimizer to classify similar agents to reduce computational complexity, which indeed improve efficiency. However, this may overlook differences between individuals, so the reason why this method can preserve the distinctive characteristics of the agents, as show in the experimental part, should be further explained in the method section。
3.	Figure 2 illustrates the principle of Group-and-Distill Meta-Prompt Optimizer. However, it seems difficult to follow. It is more intuitive to add an example to explain how IPL works.
4.	There lacks explanation for the reason why the proposed method IPL is superior to conventional prototype learning. Moreover, the principles for setting the value of M and T in IPL should be further illustrated.
5.	Experimental part: the authors should add an explanation of the indicators including JSD, T1 and bold the important data in Table 2 . Similarly, Table 3 also requires revision. The metrics of RMSE of New York和San Franciscoin are not displayed in Table 3, which seems a little bit confusing, the authors need to provide explanations.
6.	The authors should pay attention to the standardization of citations throughout the paper, especially in introduction and related works section. For example, “conventional prototype learning methods...”(line305), “the baseline represents the simulation time without optimization” (line 389), “we analyze the performance of the Generative Agent and EPR Agent ”(line 450).
7.	The authors should carefully proofread the manuscript for typos and formatting issues. There exists some typos: in the abstract: “we deisgn a “group-anddistill” prompt optimization”, “where τqα is is the proportion”(line 687) , etc.

**Questions:**

1.	Why did the authors conduct additional assessment on merely two cities: New York and Paris using the GPT-4o model in Table 2? Rather than conducting experiments in all six cities like 4o-mini?
2.	As for the experimental setup, do the following parameters: exploration rate ρ= 0.6, exploration-return trade-off parameter γ = 0.21, waiting time distribution parameters τ = 17 affect the results？
3.	In line 389，what does baseline mean? As the citation is missing, the reviewer guess whether it means the method in Park et al. (2023)？If not, comparative experiments on the Park et al. (2023) method should be added.
4.	 Why the result of baseline method is 50s/agent when the number of agents is very small in Figure 3, such as merely a single agent?

---

> ### Author Response · Authors · 2024-11-24
> **Response to Reviewer bSG3 (Part1)**
>
> Thanks for your valuable feedback, we provide the following response for your concerns:
>
> ### Weakness
> **W1. Fail to convey the motivation and novelty.**
>
> **Response:** Than you for your suggestions. We will revise this section to emphasize the motivation behind OpenCity and its novelty. Specifically, we aim to highlight the scalability challenges of LLM-based simulations and how the proposed system-level LLM request scheduler and prompt-level "Group-and-Distill" strategy address these challenges. We will also expand the related works section to provide a more comprehensive review of previous LLM-agent-based simulation platforms, clearly positioning our contributions within this context.
>
> **W2. This paper utilizes Group-and-Distill Meta-Prompt Optimizer to classify similar agents to reduce computational complexity, which indeed improve efficiency. However, this may overlook differences between individuals, so the reason why this method can preserve the distinctive characteristics of the agents, as show in the experimental part, should be further explained in the method section.**
>
> **Response:** Thank you for your valuable suggestion.  In response to your question, the Group-and-Distill method first aggregates agents based on their static attribute information, such as predefined roles or demographics. These attributes remain constant throughout the simulation and allow us to group agents effectively. By doing so, we reduce computational overhead while maintaining the individuality of agents. Each group is then represented with shared descriptive information.  During runtime, when LLM requests are made by agents within the same group, a meta-prompt is constructed (the detailed process is provided in the appendix), and Group-and-Distill extracts the descriptive information of the group as a shared context, while the personalized attributes of each agent are expressed as “situations” to retain the impact of agent individuality on the results.
>
> Empirically, as shown in Section 5.2, we demonstrate the effectiveness of this method in preserving agent individuality.  In our revision, we will expand Section 4.2 to include these detailed explanations, alongside concrete examples. We will also reference related works to situate our approach in context. Thank you again for this valuable feedback.

---

> ### Author Response · Authors · 2024-11-24
> **Response to Reviewer bSG3 (Part2)**
>
> **W3. Figure 2 illustrates the principle of Group-and-Distill Meta-Prompt Optimizer. However, it seems difficult to follow. It is more intuitive to add an example to explain how IPL works.**
>
> **Response:** Thank you for your valuable suggestion. As you mentioned, adding examples would indeed help readers better understand the mechanism of the Group-and-Distill method. The example will primarily include two parts: the input and output of the IPL process, as well as the input and output of the meta-prompt construction. We will include this detailed example in the supplementary materials of the revised version to clarify the method further. Below is an IPL example (10,000 agents in San Francisco):
>
> + **IPL Input data**
>     - **A list of agents' static attributes**, including gender, education, consumption, and occupation. One example:
>
> ```json
> {
>     "genderDescription": "male",
>     "educationDescription": "Doctoral degree",
>     "consumptionDescription": "slightly_high",
>     "occupationDescription": "IT Engineer",
> },
> ```
>
>     - $ M = 1000 $, which means that the stage one includes 1,000 agents.
>     - $ IG=10 $, which controls the number of categories in Stage One.
>     - $ T=0.5 $, which controls the threshold for category an agent to exist groups in stage two.
> + **Stage One:**
>     - In this stage, randomly pick $ M $agents and categorize them into groups.
>     - The prompt used in Stage One:
>
> ```plain
> ## Function
> According to your understanding, divide {M} enties groups. Each entry can be represents a personal information.
>
> ## Requirement
> 1. Totally {IG} groups.
> 2. Grouping should be based on maximizing differences in daily behavior between groups.
> 3. The grouping should be as general as possible to distinguish between different groups.
> 4. Each group has a descriptive name.
> 5. Each entry can only be assigned to the one and only group.
>
> ## Output Format
> The output is required to be a dictionary in JSON format, containing 2 keys.
> The first key is 'group', which is a dictonary. It contains {IG} inner keys, representing the group name, and the inner value is a list which contains all ids(int) belonging to the group.
> The second key is 'trait', which  is a dictionry. It contains {IG} inner keys, representing the group name, and the inner value is a summary of the characteristics of the group.
> Do not include any explanatory information in the output.
>
> ## Input
> {profile_list}
>
> Please give your answer:
> ```
>   - An example output of Stage One:
>
> ```json
> [
>   {
>     "High Consumption Professionals": [10, 23, 49, 52, 31, ...],
>     "trait": "Individuals with high educational backgrounds engaged in high-paying professions with high consumption habits.",
>   },
>   {
>     "Median Consumption Investors": [44, 98, ...],
>     "trait": "Professionals with median education and consumption levels primarily in investment roles.",
>   },
>   ...(total IG groups)
> ]
> ```
>
> + **Stage Two:**
>     - In this stage, sequentially process remaining agents to either categorize it into a exsit group or create a group.
>     - There are tow prompts used in Stage Two, one for grading, and the other one for create new group:
>
> ```plain
> ## Function
> You are required to give a list of grades for the input entry to support the classification of the input entry.
>
> ## Group Traits
> {group_des}
>
> ## Requirement
> 1. You should give the grade according to the relation between input entry and group trait.
> 2. The grade is between [0, 1], bigger grade means closer relation.
>
> ## Output Format
> Please output in JSON format with only a list in it.
> Example output:
> ```json
>     [x, x, x, x, ...]
> ```
> ```
>
> ```json
> ## Function
> According to your analysis, the possibility of this entry belonging to those above groups is not high, please give some insights.
>
> ## Output Format
> JSON format of dictionary, which contains 2 keys ['group_name', 'trait']. group_name is the name of this new group. 'trait' is a summary of this group.
>
> Please give your answer:
> ```
>
>   - For a specifc agent, if the output is:
>
> ```json
> [0.3, 0.6, 0.7, 0.3, 0.1, 0.1, 0.2, 0.4, 0.9, 0.4]
> ```
>
>    - then we index the group with the maximum grades and compare it with $ T $. In this case, the grade is bigger than threshold, then the agent will be categoried into group 9.
>    - In other case, if the maximum grade is lower than threshold, then it will trigger the second prompt. An example output is as follow:
>
> ```json
> {
>   "group_name": "Slightly High Consumption IT Engineers",
>   "trait": "IT engineers with slightly high consumption habits, generally holding higher degrees."
> }
> ```
>
>    - When ever a new group has been created, following LLM requests will include this information.
>    - Finally, each agent will be categorized into a group.
>    - For more specific IPL results, please refer to the directory in  [https://anonymous.4open.science/r/Anonymous-OpenCity-42BD/OpenCity](https://anonymous.4open.science/r/Anonymous-OpenCity-42BD/OpenCity).

---

> ### Author Response · Authors · 2024-11-24
> **Response to Reviewer bSG3 (Part3)**
>
> Below is an example of the input and output in Meta-Prompt Construction:
>
> ```plain
> Function:
> Decide where to go and how much time to spend.
>
> <commentblockmarker>###</commentblockmarker>
> Variables:
> !<INPUT 0>! -- Commonset
> !<INPUT 1>! -- Daily_Plan
> !<INPUT 2>! -- Current_Time
> !<INPUT 3>! -- Surrounding_Places
>
> <commentblockmarker>###</commentblockmarker>
> !<INPUT 0>!;
> Your daily plan is !<INPUT 1>!
> Now time is !<INPUT 2>!, and you have perceived these places you can go: !<INPUT 3>!;
> What's the next arrangement just now? Which place will you go next, specifically the name? And how long will you stay there, with exactly hours and minutes, always more than 1 hour. (If you just stay here, you should also answer the current place and stay time.) Please output the only answer and explain your reasons for your choice. Answer in the json format and keys are ["reason", "arrangement", "next_place", "hours", "minutes"].
> ```
>
> ```plain
> ## Shared Background:
> Commonset: !<INPUT 0>!
> Daily_Plan:
> !<INPUT 1>!
> Current_Time: !<INPUT 2>!
>
> ## Requirement:
> 1. What's the next arrangement considering your plan and current time? Which place go next? How long to spend on it, with exactly hours and minutes, always more than 1 hour. (If you just stay, you should also answer the current place and stay time.).
> 2. Consider multiple situations and give answers to each situation.
> 3. Please output a list in JSON format without other content. For each situation give a dict with keys: [\"arrangement\", \"next_place\", \"hours\", \"minutes\"].
>
> ## Situations:
> !<INPUT 3>!
> ```
>
> The prompt shown above is used in mobility simulation tasks based on the Generative Agent framework to select the next location. The prompt contains four variables: commenset (static attribute information of the agent), Daily_Plan (schedule and plans), Current_Time (current time), and Surrounding_Places (information about the surrounding environment). Among these, Surrounding_Places is dynamic, as the surrounding environment observed by agents varies depending on their location.
>
> Before optimization, the presence of dynamic variables in the prompt required each agent to independently issue an LLM request. In contrast, OpenCity’s approach aggregates static information from multiple agents (Shared Background) while retaining dynamic information (Situations), enabling the consolidation of multiple LLM requests into a single one. OpenCity automatically collects dynamic information from multiple agents at runtime and integrates it into the prompt. For detailed implementation, refer to: [https://anonymous.4open.science/r/Anonymous-OpenCity-42BD/OpenCity/OpenCity.ipynb](https://anonymous.4open.science/r/Anonymous-OpenCity-42BD/OpenCity/OpenCity.ipynb)
>
> **W4. There lacks explanation for the reason why the proposed method IPL is superior to conventional prototype learning. Moreover, the principles for setting the value of M and T in IPL should be further illustrated.**
>
> **Response:** Thank you for your insightful feedback. In comparison to conventional prototype learning methods, IPL provides significant advantages, especially when applied to large-scale LLM scenarios. Traditional prototype learning approaches often rely on training models within a fixed parameter space, which can be computationally intensive and risk sacrificing the model's generalization ability. This limitation becomes even more pronounced when handling diverse and complex datasets, such as urban residents with attributes like "profession" or "consumption level," which must first be converted into a numerical format. In contrast, IPL leverages the in-context learning capabilities inherent in large language models, bypassing the need for additional training. By utilizing the LLM's built-in semantic understanding and reasoning, IPL ensures greater efficiency and preserves the LLM’s generalization ability. Furthermore, IPL integrates seamlessly with the operational mechanisms of LLM agents, ensuring consistency between the meta-learning process and agent behaviors, which enhances overall adaptability and scalability.
>
> Regarding the parameters M and T:  The M hyperparameter controls the first stage of IPL, while T influences the second stage. Specifically, the M parameter defines the initial reference entries for meta-learning (for example, with 10,000 LLM agents, when M is set to 1,000, agents are randomly selected for an initial grouping). This parameter is constrained by the LLM’s context length and its ability to process long texts. The T parameter is the grouping threshold: the larger the T value, the finer the granularity of the groups and the higher the number of groups. Conversely, a smaller T results in fewer groups.
>
>  We will provide a detailed explanation of these advantages and parameters in the revised version.

---

> ### Author Response · Authors · 2024-11-24
> **Response to Reviewer bSG3 (Part4)**
>
> **W5. Experiment part details**
>
> + **Explanation of the indicators, including JSD and T1**: Specifically, JSD measures the similarity between distributions, while T1 captures the accuracy of key selections in the simulation.
> + **Bold the important data in the results table**: We appreciate your suggestion to bold key results in the tables, which we will implement to improve readability.
> + **About the missing **$ R_{MSE} $** of New York and San Francisco**: The absence of $ R_{MSE} $ for New York and San Francisco is due to the nature of the dataset in these cities, which aggregates user data without individual trajectories. Since $ R_{MSE} $ measures the difference in radius of gyration between real and simulated users, it is not applicable in these cases. We will clarify this in the manuscript.
>
> **W6. Citation and reference issues**
>
> **Response：** We will thoroughly review and standardize all citations throughout the paper. Ambiguities, such as those in lines 305 and 389, will be resolved, and all references will adhere to proper formatting guidelines.
>
> **W7. Manuscript for typos and formatting issues**
>
> **Response:** We acknowledge the presence of typos and formatting issues. We will carefully proofread the manuscript to eliminate errors such as "deisgn" (abstract) and "is is" (line 687).
>
> ### Question
> **Q1. Why did the authors conduct additional assessment on merely two cities: New York and Paris using the GPT-4o model in Table 2? Rather than conducting experiments in all six cities like 4o-mini?**
>
> **Response:** We chose New York and Paris as representative cities because they capture diverse urban dynamics. The results in the two cities have shown consistency with GPT-4o and 4o-mini, which means that our conclusion can be generalized to other cities. What's more, GPT-4o is significantly more expensive than GPT-4o-mini, which limits the scope of experiments.  We will clarify this in the experimental setup.
>
> **Q2. As for the experimental setup, do the following parameters: exploration rate ρ= 0.6, exploration-return trade-off parameter γ = 0.21, waiting time distribution parameters τ = 17 affect the results？**
>
> **Response:** The parameters (ρ=0.6,γ=0.21,τ=17) are specific to the EPR (Explore and Preferential Return) model, which is just a baseline in urban dynamics simulation part. While these parameters influence EPR outcomes, they were sourced from statistical insights into mobility patterns  Song et al. (2010). Using these values ensures the results align with empirical human behavior.
>
> Song, C.; Koren, T.; Wang, P.; Barabási, A.-L. Modelling the Scaling Properties of Human Mobility. _Nature Phys_ **2010**, _6_ (10), 818–823. [https://doi.org/10.1038/nphys1760](https://doi.org/10.1038/nphys1760).
>
> **Q3. In line 389，what does baseline mean? As the citation is missing, the reviewer guess whether it means the method in Park et al. (2023)？If not, comparative experiments on the Park et al. (2023) method should be added.**
>
> **Response:** The term "baseline" refers to the generative agents framework introduced by Park et al. (2023) **without our acceleration methods**. We will explicitly cite this work and clarify the distinction in the manuscript. Comparative experiments with this baseline are already included to demonstrate the advantages of our proposed approach.
>
> **Q4. Why the result of baseline method is 50s/agent when the number of agents is very small in Figure 3, such as merely a single agent?**
>
> **Response:** The baseline method exhibits a consistent per-agent time of 50s because it does not leverage parallelism or optimization techniques. Our acceleration methods, such as system-level LLM request scheduler and prompt-level "group-and-distill" strategy,  become increasingly effective as the number of agents grows. This is due to greater opportunities for agent state reuse and prompt simplification.  For a single agent, these optimizations have limited impact, which accounts for the observed result.

---

> > ### Comment · Reviewer_bSG3 · 2024-11-25
> > **Thank the authors for their detailed feedback**
> >
> > After reading the rebuttal as well as other reviewers’ comments, I keep my score.

---

### Official Review · Reviewer_2bbN · 2024-11-03

**Soundness:** 1
**Presentation:** 2
**Contribution:** 1
**Rating:** 3
**Confidence:** 4

**Summary:**

The paper describes an approach	where LLM agents are used to simulate individual behaviour in large (city-scale) simulations of people.	The proposed platform uses LLM agents that can adapt their behaviour depending on context and memory. This is different to the classic agent based approach for this type of simulation where behaviours are static over time.
The development of the platform is one of the main contributions of the work, and the development of a user-friendly web interface is another contribution highlighted by the authors. From a machine learning perspective, the proposed “group-and-distill” approach to reduce LLM usage is the main contribution of the work, essentially a clustering approach before prompting the LLM for each cluster (as opposed to prompting an LLM for each individual).

**Strengths:**

The use of a LLM for the purpose of larger scale population modelling appears to be novel, and the suggested group-and-distill approach enables this idea, with relatively low hardware resources.
Overall, considerable effort appears to have gone into development of the system. The system could be an interesting resource for research in complex systems.

**Weaknesses:**

The paper has quite a broad focus, like an overall project report. For a venue like iclr, it would have been better to focus on the specific contributions in machine learning, and to provide more technical details rather than an overall description of architecture and usability aspects as the main contributions. In its current form, ICLR does not appear to be the right venue for the work as it is presented.

The work lacks depths in aspects that I would see essential for any ML paper: for example the group-and-distill concept is introduced, but the paper is very sparse in detail of the specific algorithms. Similarly it would have been interesting to see what are the initial prompts and the optimised prompts, in contrast.
Any details comparing to the original approach without group-and-distill / ablation would have been an improvement too.

Moreover it didn’t become clear to me what LLM has been used or how was it trained, and how do LLM outputs influence agents’ behaviours.

Finally, the paper mentioned at the beginning the explainability of ABM as an advantage over black box neural network approaches. with the lack of detail on how the actions are influenced by the LLM or how the LLM are trained or fine tuned, the proposed model has the same disadvantage as any other neural network model.

Minor presentation issues:

"Agent-based models (ABMs) were first introduced to urban studies in the seminal work of Thomas Schelling about 50 years ago Schelling (2006)"
- if the work referenced was from approx 50 years ago, Schelling 2006 is the wrong reference. I believe the correct year would be 1978.
- there are two	kinds of citations, narrative (like the one in the sentence), and parenthetical (Schelling, 2006). It doesn't make sense to use	narrative style	when it doesn't fit into the sentence structure. In LaTeX with natbib, this is the difference between \citet and \citep.
The referencing is an issue throughout the paper.

**Questions:**

While the approach and system are interesting, I don't see this paper as a good fit for ICLR, in its present form, and suggest it be rejected.
Some of my questions:

- What LLM model is used in the simulation, and how was it trained?
- How do the prompts look before and after applying the group-and-distill approach?
- What is the output of the LLM, and how does it influence agent behaviour?
- To what extent is the group-and-distill technique generalisable beyond urban simulations?
- How does the platform maintain long-term consistency in agent behaviour, given the variability in LLM responses?
- What mechanisms are in place to manage or correct for inconsistencies in agent behaviour across prompts?
- Is there an evaluation of the platform’s accuracy in simulating real-world behaviours compared to traditional ABMs?

---

> ### Author Response · Authors · 2024-11-24
> **Response to Reviewer 2bbN (Part1)**
>
> Thank you for your valuable reviews.  Here I'd like to answer your questions as follows:
>
> ### Weaknesses
> **W1: Lack of focus on specific machine learning contributions**
>
> **Response:** Our primary objective is to propose a scalable platform that integrates the important trend of generative agents in simulating human societies. It has emerged as an important research areas to unleash the power of LLMs to simulate and understand human societies[1][2]. A critical research gap for this area is the scalability of LLM agents. Our work seeks to address this specific and important problem. It makes specific contribution to benchmarking generative agents. What's more, while we understand that ICLR emphasizes specific ML innovations, we note that platform or benchmark-focused work has been accepted in prior conferences[3][4].
>
> [1]Park, Joon Sung, et al. "Generative agents: Interactive simulacra of human behavior." Proceedings of the 36th annual acm symposium on user interface software and technology. 2023.
>
> [2] Li, Nian, et al. "Econagent: large language model-empowered agents for simulating macroeconomic activities." Proceedings of the 62nd Annual Meeting of the Association for Computational Linguistics (Volume 1: Long Papers). 2024.
>
> [3] Huang, S., Weng, J., Charakorn, R., Lin, M., Xu, Z., & Ontañón, S. (2023, September). Cleanba: A Reproducible and Efficient Distributed Reinforcement Learning Platform. In The Twelfth International Conference on Learning Representations. [https://openreview.net/forum?id=Diq6urt3lS](https://openreview.net/forum?id=Diq6urt3lS)
>
> [4] Huang, Y., Shi, J., Li, Y., Fan, C., Wu, S., Zhang, Q., ... & Sun, L. (2023). Metatool benchmark for large language models: Deciding whether to use tools and which to use. arXiv preprint arXiv:2310.03128. [https://arxiv.org/pdf/2310.03128](https://arxiv.org/pdf/2310.03128)
>
> **W2: Insufficient technical depth**
>
> **Response:**
>
> Our technical contributions include system-level LLM request scheduler and prompt-level “group-and-distill”strategy  making the platform scalable.  There may be some missing details in the current version, and we will address this by expanding the **supplementary materials**. These will include:
>
> + **Prompt examples** throughout the process
> + Detailed descriptions of the **group-and-distill** methodology
>
> Below is an example of the initial prompt and corresponding optimized prompt:
>
> ```plain
> Function:
> Decide where to go and how much time to spend.
>
> <commentblockmarker>###</commentblockmarker>
> Variables:
> !<INPUT 0>! -- Commonset
> !<INPUT 1>! -- Daily_Plan
> !<INPUT 2>! -- Current_Time
> !<INPUT 3>! -- Surrounding_Places
>
> <commentblockmarker>###</commentblockmarker>
> !<INPUT 0>!;
> Your daily plan is !<INPUT 1>!
> Now time is !<INPUT 2>!, and you have perceived these places you can go: !<INPUT 3>!;
> What's the next arrangement just now? Which place will you go next, specifically the name? And how long will you stay there, with exactly hours and minutes, always more than 1 hour. (If you just stay here, you should also answer the current place and stay time.) Please output the only answer and explain your reasons for your choice. Answer in the json format and keys are ["reason", "arrangement", "next_place", "hours", "minutes"].
> ```
>
> ```plain
> ## Shared Background:
> Commonset: !<INPUT 0>!
> Daily_Plan:
> !<INPUT 1>!
> Current_Time: !<INPUT 2>!
>
> ## Requirement:
> 1. What's the next arrangement considering your plan and current time? Which place go next? How long to spend on it, with exactly hours and minutes, always more than 1 hour. (If you just stay, you should also answer the current place and stay time.).
> 2. Consider multiple situations and give answers to each situation.
> 3. Please output a list in JSON format without other content. For each situation give a dict with keys: [\"arrangement\", \"next_place\", \"hours\", \"minutes\"].
>
> ## Situations:
> !<INPUT 3>!
> ```
>
> The prompt shown above is used in mobility simulation tasks based on the Generative Agent framework to select the next location. The prompt contains four variables: commenset (static attribute information of the agent), Daily_Plan (schedule and plans), Current_Time (current time), and Surrounding_Places (information about the surrounding environment). Among these, Surrounding_Places is dynamic, as the surrounding environment observed by agents varies depending on their location.
>
> Before optimization, the presence of dynamic variables in the prompt required each agent to independently issue an LLM request. In contrast, OpenCity’s approach aggregates static information from multiple agents (Shared Background) while retaining dynamic information (Situations), enabling the consolidation of multiple LLM requests into a single one. For detailed implementation, refer to: [https://anonymous.4open.science/r/Anonymous-OpenCity-42BD/OpenCity/OpenCity.ipynb](https://anonymous.4open.science/r/Anonymous-OpenCity-42BD/OpenCity/OpenCity.ipynb)

---

> ### Author Response · Authors · 2024-11-24
> **Response to Reviewer 2bbN (Part2)**
>
> **W3: Details on LLM usage and agent behavior**
> **Response:** In our current implementation, we leverage commercial APIs for GPT-4o and GPT-4o-mini without additional training. Agents’ behaviors are directly determined by the LLM outputs, which include planned actions, thoughts on perceived environments, and the next movement location. These outputs form the foundation of the agents' decisions and mobility behavior.
>
> **W4: Explainability of the model**
> **Response:** One advantage of our platform is the ability to examine an agent’s "mind" by analyzing its prompts and the corresponding LLM responses. This transparency allows researchers to trace decision-making processes. We acknowledge that this was insufficiently detailed in the submission and will include examples in the revised version to clarify.
>
> **W5: Citation and reference issues**
> **Response:** Thank you for pointing out the reference inaccuracies and citation style inconsistencies. We will correct these issues and ensure the format adheres to standard conventions.
>
> ### Questions
> **Q1: What LLM model is used in the simulation, and how was it trained?**
> Response: The LLMs used in our platform are GPT-4o and GPT-4o-mini, accessed via the OpenAI API. These models are not trained or fine-tuned in this work, as our focus is on leveraging their reasoning and role-playing capabilities within the ABM framework.
>
> **Q2 : How do the prompts look before and after applying the group-and-distill approach?**
> Response: See prompt example in the **W2 Response.**
>
> **Q3: What is the output of the LLM, and how does it influence agent behavior?**
> Response: The outputs include agents' daily plans, their interpretation of the environment, and subsequent actions. In our case, the key output is the next location that an agent moves to, which drives its mobility behavior. These outputs influence agents' decisions and interactions within the simulation.
>
> **Q4: To what extent is the group-and-distill technique generalizable beyond urban simulations?**
> **Response:** The group-and-distill approach is inherently generalizable. Its core principle，sharing common contextual information among agents to reduce redundancy，can be applied to any simulation that involves coordination among large numbers of LLM-based agents, whether the agents have both static and dynamic attributes, or exhibit spatiotemporal dynamics.
>
> **Q5 : How does the platform handle variability in LLM responses to maintain consistency in agent behavior?**
> **Response: **Consistency is maintained by employing memory mechanisms and recurrent context updates, which is an established framework in generative agents[1]. Each agent retains a structured history of interactions and observations, ensuring coherent and contextually appropriate decisions over time. This is directly related to the agent design and the prompt organization method.
>
> Furthermore, the dynamic nature of the LLM agents themselves mirrors human behavior characteristics. Even when facing the same situation—such as the same time and location when choosing a dining place—humans might make different choices, reflecting the inherent variability and adaptability in decision-making. This dynamic capability adds a layer of realism to agent behavior, allowing for more flexible and human-like interactions.
>
> **Q6: What mechanisms are in place to manage or correct inconsistencies in agent behavior across prompts?**
> **Response: **The memory mechanisms are employed where agents past outputs and actions inform their future prompts. Additionally, clustering similar agents into groups helps reduce variability at scale. These measures help mitigate inconsistencies and promote stable behaviors across simulations.
>
> **Q7: Is there an evaluation of the platform’s accuracy in simulating real-world behaviors compared to traditional ABMs?**
> **Response:** We have this evaluation in Section 5.3 Table 3, where we compare the LLM-driven generative agents and the traditional ABMs EPR models. The results show that the LLM agent performs as well as or better than the classical rule-based EPR agent .
>
> We appreciate your constructive feedback, which will help us improve both our work and its presentation. We hope these clarifications address your concerns and demonstrate the value and potential impact of our platform. Thank you again for your insightful review.

---

> > ### Comment · Reviewer_2bbN · 2024-11-26
> >
> > Thank you for the detailed responses, and the work spent on improving the paper.
> > The answers do address my questions, though the lack of technical detail is only marginally resolved. I do understand the question of scalability is important, I am just not sure ICLR is the best / right venue for the work presented, not a conference on learning representations (I would have expected the work to appear at eg AAMAS). I would feel more confident if the specific technical contribution that enables the scalability was a main part of the paper, or if indeed a shared representation was learnt.

---

### Meta-Review · Area_Chair_3RvB · 2024-12-20

**Metareview:**

The paper proposes OpenCity, a platform combining agent-based modeling (ABM) with large language models (LLMs) to simulate urban dynamics at scale. By leveraging techniques like the “group-and-distill” prompt optimization, the platform achieves significant computational efficiencies, including a 600-fold speedup and reduced resource consumption in large-scale simulations. The paper claims to establish a new benchmark for LLM-powered urban simulation and provides a user-friendly web interface for broader accessibility.

While the proposed OpenCity model shows promising results, the reviewers have identified several weaknesses that need to be addressed:
1. The paper’s broad focus, which includes platform architecture and usability aspects, lacks sufficient technical depth on the machine learning contributions, especially regarding the “group-and-distill” approach.
2. The absence of detailed algorithmic descriptions and comparisons, such as ablation studies or specifics on optimized prompts, weakens the paper’s technical rigor.
3. There is a lack of clarity on which LLM model is used, how it is trained, and how its outputs influence agent behavior, leaving the model’s inner workings inadequately explained.

Based on these weaknesses, we recommend rejecting this paper. We hope this feedback helps the authors improve their paper.

**Additional Comments On Reviewer Discussion:**

In their rebuttal, the authors made several improvements, including clarifications and updates to the presentation, which help reviewers better understand the contributions of the paper. However, the reviewers’ concerns regarding the technical depth, particularly the lack of detailed comparisons and algorithmic explanations, remain unresolved. As a result, I recommend rejection based on the reviewers’ feedback.

---

### Decision · Program_Chairs · 2025-01-22

Reject